# SHORTCUTS IN MATERIAL DESIGN: EFFICIENT GENERATIVE MODELING OF AMORPHOUS MATERIALS

## ABSTRACT

Amorphous materials, such as glasses, are solids that lack long-range atomic order but possess complex short- and medium-range order. Inverse design of amorphous materials with probabilistic generative models aims to generate the atomic positions and elements of amorphous materials given desired properties. It has emerged as a promising approach for facilitating the application of amorphous materials in domains such as energy storage and thermal management. In this paper, we introduce MDShortcut, an inference- and training-efficient probabilistic generative model for amorphous materials. MDShortcut enables accurate inference of diverse short- and medium-range structures in amorphous materials with only a few sampling steps, mitigating the need for an excessive number of sampling steps that hinders inference efficiency. MDShortcut can be trained once with all relevant properties and perform inference conditioned on arbitrary combinations of desired properties, mitigating the need for training one model for each combination. Experiments on two amorphous materials datasets with diverse structures and properties demonstrate that MDShortcut achieves its design goals.

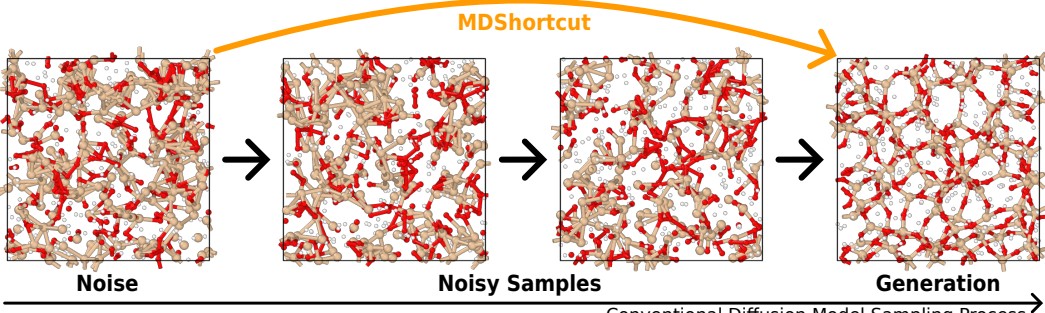

Figure 1: MDShortcut generates structurally accurate amorphous material samples with one or few sampling steps, enabling high-throughput inverse design of amorphous materials.

## 1    INTRODUCTION

Glasses and other amorphous materials are solids that lack a periodic atomic arrangement or long-range atomic order, yet exhibit complex short- and medium-range order. In other words, their atoms are randomly arranged overall but still form organized clusters in localized regions. They have shown great potential in diverse domains, including energy storage, thermal management, and advanced materials (Liu et al., 2024). To advance the design of amorphous materials with desired properties, inverse design has emerged as a promising approach for taking *shortcuts*, instead of relying on resource-intensive trial-and-error processes. It starts with target properties and works backward to determine the necessary atomic configurations. One intuitive way to implement this approach is through probabilistic generative models (Kingma & Welling, 2014; Goodfellow et al., 2014), especially those based on diffusion models (Ho et al., 2020), which generate atomic positions and elements conditioned on desired properties by transforming random noise to targets through a multi-step Markov process (Figure 1). Such models have shown success in generating relatively

small-scale atomic systems, including crystalline materials and molecules (Wu et al., 2022; Xie et al., 2022; Hoogeboom et al., 2022; Zeni et al., 2025), but remain under-developed for amorphous materials due to the lack of large-scale datasets and their unique atomic ordering characteristics (Yang & Schwalbe-Koda, 2025; Finkler et al., 2025).

In this paper, we focus on the **inference and training efficiency** of probabilistic generative models for amorphous materials. Specifically, *inference efficiency* is hindered by the need for many sampling steps to accurately generate structures in amorphous materials. Due to their lack of long-range atomic order, amorphous material samples are usually represented in large periodic cells with diverse short- and medium-range orders. We show that accurate generation of such structures requires large numbers of diffusion sampling steps to explore optimal atomic positions, for both state-of-the-art generative modeling frameworks: score-matching SDEs (Song et al., 2021) and flow-matching ODEs (Lipman et al., 2023). On the other hand, *training efficiency* is hindered by the variety of properties on which the generative models need to condition. In practice, inverse design of amorphous materials focus on different sets of properties to fit in different needs, yet training one model for each set of properties necessitates training and maintaining numerous models. Techniques such as classifier guidance (Dhariwal & Nichol, 2021; Lin et al., 2025) enables training of one uniform unconditioned generative model and guide the model's generation with dedicated classifiers. Yet, the poor availability of differentiable classifiers for amorphous materials can make implementing such techniques impractical.

To this end, we propose **MDShortcut**, a model for taking shortcuts in material design that serves as both an effective inverse design model and an efficient probabilistic generative model for amorphous materials. We build a *material differential equation* framework as the foundation for generative modeling of amorphous materials, which generates material samples starting from random noise and gradually removes noise from atomic positions and elements until a noise-free target sample is reached. We derive two baseline models, material SDE and material ODE, and MDShortcut from this framework. Both baselines are found to be capable of generating structurally accurate amorphous material samples, but at the cost of many sampling steps. Inspired by recent efforts in one-step diffusion models (Frans et al., 2025; Geng et al., 2025), MDShortcut learns shortcuts that properly jump between large step sizes, enabling it to perform generation in few steps. We demonstrate that compared to the baselines, MDShortcut can reduce inference time by up to 99% without compromising structural accuracy (Section 3.3). We also introduce a *flexible material denoiser* as the core learnable network of MDShortcut. This denoiser can be trained once conditioned on all relevant properties of amorphous materials and used for inference conditioned on arbitrary subsets of desired properties. Properties absent during inference are represented as "null properties", which is equivalent to being unconditioned on these properties. The denoiser utilizes a non-learning approach to calculating representations for null properties (Sadat et al., 2025), which mitigates the need for training dedicated unconditioned model or classifiers, further improving the training efficiency. We show in experiments that inference partially conditioned on a subset of properties closely matches inference with a denoiser specifically trained for these properties (Section 3.4).

Finally, we utilize two amorphous material datasets (Finkler et al., 2025) for experimental evaluation: a single-element amorphous Silicon (a-Si) dataset for evaluating structural accuracy, and an amorphous Silica (a-SiO$_2$) dataset for evaluating inverse design performance, whose properties are primarily determined by atomic structures and densities. Experiments on these datasets provide evidence that MDShortcut achieves its design goals.

## 2 MDShortcut

### 2.1 Material Differential Equation

**Representation of amorphous materials.** We represent an amorphous material sample as the positions and elements of atoms inside a periodic cell, formally a tuple $\mathcal{M} = (\boldsymbol{C}, \boldsymbol{X}, \boldsymbol{E})$ of three matrices. $\boldsymbol{C} \in \mathbb{R}^{3 \times 3}$ contains the three lattice vectors of the cell, $\boldsymbol{X} \in \mathbb{R}^{n_a \times 3}$ represents the positions of $n_a$ atoms, and $\boldsymbol{E} \in \mathbb{R}^{n_a \times d_E}$ contains the one-hot embeddings of atomic elements, where $d_E$ is the total number of elements under consideration. We also represent the set of $n_p$ relevant properties as $\boldsymbol{y} \in \mathbb{R}^{n_p}$, where each value denotes the magnitude of that property.

Note that we incorporate "ghost atoms", a special atom type, into each material sample so that the generative model can control the density of the sample without modifying $C$ or the number of atoms. This approach enables generating materials with specific density targets while maintaining a fixed total number of atoms and preserving the model's equivariant architecture. In the datasets, these atoms are randomly positioned into the cell so that the total number of atoms $n_a = \lfloor \rho \cdot \text{Vol}(C) \rfloor$, where $\rho$ is the maximum density. During training and inference, ghost atoms are treated like normal atoms but are assigned a special chemical element class. The model can adjust the density of the sample by changing the fraction of atoms that are assigned the ghost atom type, and as a final step after generation, ghost atoms are removed from the sample.

**Generative modeling of amorphous materials.** Our goal is to sample from the distribution of samples conditioned on properties, i.e., $p(\mathcal{M}|\boldsymbol{y})$, to generate new samples with desired properties. Since the exact form of this distribution is unknown, we parameterize the sampling process as a time-dependent differential equation, which we term *material differential equation*:

$$d\mathcal{M}_t = \mu(\mathcal{M}_t, \boldsymbol{y}, t)dt + \sigma(t)dW_t, \quad t \in [1, 0] \tag{1}$$

which defines a continuous-time process that transforms a noise sample $\mathcal{M}_1$ that can be easily sampled from a prior distribution to a target sample $\mathcal{M}_0$. $\mu(\mathcal{M}_t, \boldsymbol{y}, t)$ is the deterministic drift coefficient and $\sigma(t)$ is the diffusion coefficient controlling the magnitude of the stochastic Wiener process $W_t$. In practice, the sampling process is applied to positions $\boldsymbol{X}$ and element embeddings $\boldsymbol{E}$ with the cell $C$ unchanged, and both $\mu$ and $\sigma$ have two components for positions and element embeddings respectively, which we denote as $\mu_{\boldsymbol{X}}, \mu_{\boldsymbol{E}}, \sigma_{\boldsymbol{X}},$ and $\sigma_{\boldsymbol{E}}$.

To generate a sample $\mathcal{M}_0$, we sample noise $\mathcal{M}_1$ and solve the above differential equation using the Euler-Maruyama method, where we discretize the time span $[1, 0]$ into $n_s$ steps with step size $\Delta t = 1/n_s$. Each step is performed as:

$$\mathcal{M}_{t-\Delta t} = \mathcal{M}_t - \Delta t\mu(\mathcal{M}_t, \boldsymbol{y}, t) + \sqrt{\Delta t}\sigma(t)\epsilon, \quad \epsilon \sim \mathcal{N}(0, 1) \tag{2}$$

This can be intuitively understood as moving the atomic positions and element embeddings of a sample in the direction and speed functions of $\mu(\mathcal{M}_t, \boldsymbol{y}, t)$, while adding a certain magnitude of noise.

**Material ODE and material SDE.** In practice, the drift coefficient $\mu$ is estimated with a learnable neural network $\mu_\theta(\mathcal{M}_t, \boldsymbol{y}, t)$ with $\theta$ being the set of learnable parameters, and the diffusion coefficient $\sigma$ is pre-defined for simplicity. The network is trained by independently sampling $\mathcal{M}_1$ from the prior distribution, sampling $\mathcal{M}_0$ from the training dataset, calculating the ground truth $\mu$ and $\mathcal{M}_t$, and supervising $\mu_\theta$ with $\mu$.

We introduce two variants of Eq. 1, material ODE and material SDE, with specific formulations of the above ground truth. The *material ODE* is the ordinary differential equation variant, which formulates the ground truth following the optimal transport flow (Lipman et al., 2023): $\sigma(t) = 0$, $\mathcal{M}_t$ is linear interpolation between $\mathcal{M}_1$ and $\mathcal{M}_0$, and $\mu$ is set to pointing from $\mathcal{M}_1$ to $\mathcal{M}_0$. The network $\mu_\theta$ directly predicts $\mu$, and is trained with $L_2$ loss, denoted as $\mathcal{L}_{\text{ODE}}$. The *material SDE* is the stochastic differential equation variant, which formulates the ground truth following the score-matching SDE (Song et al., 2021) with a variance exploding noise schedule on positions and a variance preserving noise schedule with cosine progression on element embeddings. The drift coefficient is parameterized with the scores of positions and element embeddings. The network $\mu_\theta$ predicts the noise components, which are then used to parameterize the scores. The network is trained with $L_2$ loss, denoted as $\mathcal{L}_{\text{SDE}}$. Detailed formulation of both variants are given in Appendix A.2 and A.3.

For both variants, we sample the positions $\boldsymbol{X}_1$ and element embeddings $\boldsymbol{E}_1$ of a noise sample $\mathcal{M}_1$ from uniform random distribution within the cell $C$ and standard normal distribution, respectively. Comparatively, material ODE does not have a stochastic component and should be more stable for generation with a small number $n_s$ of steps, while the stochastic component in material SDE gives it more freedom in exploring optimal structures during generation. We show that in practice both variants require many steps to generate structurally accurate amorphous material samples (Section 3.3).

## 2.2 Learning Shortcuts in Material SDE

Works on one-step diffusion models (Frans et al., 2025; Geng et al., 2025) provide insights into why material ODE and SDE require many sampling steps: the direction and speed defined by the

drift coefficient $\mu$ change rapidly over time $t$. In Eq. 2, the material sample is updated using the instantaneous drift at the current time, resulting in inaccurate generation when a large step size is used. Inspired by these works, the proposed MDShortcut is built on material SDE, but with additional step size-awareness and provides *shortcuts*, i.e., instead of updating the sample using the instantaneous drift, the model can update the sample accurately across long step sizes.

Specifically, MDShortcut trains a neural network $u_\theta(\mathcal{M}_t, \boldsymbol{y}, t, \Delta t)$ with similar architecture to $\mu_\theta$ but additionally takes the step size $\Delta t$ into consideration. A ground truth shortcut $u$ across $t$ and $t - \Delta t$ is defined as:

$$u(\mathcal{M}_t, \boldsymbol{y}, t, \Delta t) = \frac{1}{\Delta t} \int_{t-\Delta t}^t \mu(\mathcal{M}_\tau, \boldsymbol{y}, \tau)d\tau + \frac{1}{\Delta t} \int_{t-\Delta t}^t \sigma(\tau)dW_\tau \tag{3}$$

Note that similar to $\mu$, a shortcut also contains two components for positions and element embeddings denoted as $u_{\boldsymbol{X}}$ and $u_{\boldsymbol{E}}$ respectively, but we do not present the separate formulas for simplicity. The network $u_\theta$ still predicts the noise first, and the predicted shortcuts are calculated following the parameterization of $\mu$ in material SDE.

To learn the shortcuts in a computationally efficient way, we follow the idea of self-consistency loss in Frans et al. (2025) and implement the shortcut loss for material SDE:

$$\mathcal{L}_{\text{SC}} = \mathbb{E}_{\epsilon, \mathcal{M}_0, t, \Delta t} \| u_\theta(\mathcal{M}_t, \boldsymbol{y}, t, 2\Delta t) - \text{sg}(u_{\text{target}}) \|^2$$
$$u_{\text{target}} = (u_\theta(\mathcal{M}_t, \boldsymbol{y}, t, \Delta t) + u_\theta(\hat{\mathcal{M}}_{t-\Delta t}, \boldsymbol{y}, t - \Delta t, \Delta t)) \tag{4}$$
$$\hat{\mathcal{M}}_{t-\Delta t} = \mathcal{M}_t - \Delta t u_\theta(\mathcal{M}_t, \boldsymbol{y}, t, \Delta t) + \sqrt{\Delta t}\sigma(t)\epsilon$$

where sg is stop gradient. Essentially, we leverage the fact that two consecutive shortcuts should equal one shortcut with double the step size. For more stable training, when computing Eq. 4 we follow Song et al. (2021) by removing the stochastic component in material SDE and modify the drift coefficients in compensation. Detailed formulations are given in Appendix A.4.

Finally, the network $u_\theta$ is trained with the combined loss of material SDE and shortcuts: $\mathcal{L} = \mathcal{L}_{\text{SDE}} + \mathcal{L}_{\text{SC}}$, leveraging the equivalence $u_\theta(\mathcal{M}_t, \boldsymbol{y}, t, 0) \equiv \mu_\theta(\mathcal{M}_t, \boldsymbol{y}, t)$ when calculating $\mathcal{L}_{\text{SDE}}$.

## 2.3 FLEXIBLE MATERIAL DENOISER

We implement $\mu_\theta$ and $u_\theta$ as the flexible material denoiser. Both networks transform an input material sample $\mathcal{M}_t$, properties $\boldsymbol{y}$, and time $t$ into position and element embedding components, with the only difference being that $u_\theta$ additionally incorporates step size $\Delta t$:

$$\mu_\theta : (\mathcal{M}_t, \boldsymbol{y}, t) \mapsto (\hat{\mu}_{\boldsymbol{X}}, \hat{\mu}_{\boldsymbol{E}}) \quad \text{or} \quad u_\theta : (\mathcal{M}_t, \boldsymbol{y}, t, \Delta t) \mapsto (\hat{u}_{\boldsymbol{X}}, \hat{u}_{\boldsymbol{E}}) \tag{5}$$

**Flexible property embedding.** The denoiser can be trained once with all available properties $\boldsymbol{y}$ and used for inference when $\boldsymbol{y}$ is only partially available. This is achieved through the design of the property embedding layer. Specifically, inspired by Sadat et al. (2025), the embedding vector of each property $y_i$ is calculated as:

$$\boldsymbol{h}_{y_i} = \begin{cases} \text{LayerNorm}(\text{Linear}(y_i)) & \text{if } y_i \text{ is available} \\ \boldsymbol{\xi}_{\text{emb}} \sim \mathcal{N}(0, 1) & \text{if } y_i \text{ is unavailable} \end{cases} \tag{6}$$

where $\boldsymbol{h}_{y_i} \in \mathbb{R}^{d_y}$ with $d_y$ being the embedding dimension of properties, and $\boldsymbol{\xi}_{\text{emb}}$ is a null property embedding vector sampled from a standard normal distribution. Since $\boldsymbol{\xi}_{\text{emb}}$ follows the same distribution as embeddings of available properties but is sampled independently of $\mathcal{M}$, conditioning on it is equivalent to unconditional generation for unavailable properties. Compared to classifier guidance (Dhariwal & Nichol, 2021), this design does not require dedicated differentiable classifiers, which have limited availability for amorphous materials, and also face difficulties such as not always working on noisy samples. Compared to classifier-free guidance (Ho & Salimans, 2022), this design does not require special training procedures.

**E(n)-equivariant backbone.** To preserve the geometric equivariance (permutation, translation, rotation, and mirror equivariance) of amorphous material samples, we use an equivariant graph neural network (EGNN) (Satorras et al., 2021) as the backbone of the flexible material denoiser.

The input graph to EGNN is composed of atoms in each material sample where the edges are atom pairs with distance less than 6.5 Å, considering periodic boundary conditions. Each node feature is the concatenation of the element embedding of the corresponding atom, property embeddings, time $t$, and step size $\Delta t$ in the case of $u_\theta$. Each edge feature is derived from the edge length. The EGNN calculates a weight for each edge and then updates the positions and element embeddings of each atom with the weights. Implementation details of the EGNN backbone are provided in Appendix A.5.

## 3 EXPERIMENTS

We evaluate the performance of MDShortcut against the two baseline models (Material SDE and Material ODE) on two amorphous material datasets.

### 3.1 AMORPHOUS MATERIAL DATASETS

The two datasets (Finkler et al., 2025) are obtained using classical molecular dynamics simulations workflows based on LAMMPS (Thompson et al., 2022) and ASE (Larsen et al., 2017). More details about the datasets are provided in Appendix A.9.

**Amorphous Silicon (a-Si) dataset.**  The a-Si dataset contains 10,000 samples, each with 256 silicon (Si) atoms, specifically for evaluating the structural accuracy of generation. Samples are obtained using the Stillinger-Weber potential (Stillinger & Weber, 1985) from the melt at 2500 K.

**Amorphous Silica (a-SiO$_2$) dataset.**  The a-SiO$_2$ dataset contains silica (SiO$_2$) samples that vary in size (between 80 and 250 atoms) and whose properties are dependent on structures and densities, since the composition is fixed. Samples are obtained using the Tersoff potential parameterized by Munetoh et al. (2007). Samples are initially melted at 3500 K and then immediately quenched using a local geometry optimization to avoid relaxation effects. Finally, samples are equilibrated at 300 K for 10 ps.

On the a-Si dataset, since the elements and densities are fixed and no property is assigned to the samples, we are particularly interested in the structural accuracy of the samples generated by models. On the a-SiO$_2$ dataset, the inverse design performance of the models is evaluated. Specifically, we calculate the accuracy of properties of the samples generated by models against the target properties given to the models.

### 3.2 EVALUATION METRICS

**Structural metrics.**  We evaluate structural accuracy using radial distribution functions (RDF) and angular distribution functions (ADF). RDF measures the probability of finding atom pairs at various distances, capturing short- and medium-range order in amorphous materials. ADF characterizes the distribution of bond angles between atomic triplets, revealing local coordination geometries. We quantify the agreement between generated and reference structures using Root Mean Square Deviation (RMSD) of these distributions, where lower values indicate better structural accuracy. Implementation details are provided in Appendix A.6.

**Properties of material samples.**  We are interested in the following properties of samples in the a-SiO$_2$ dataset. Shear modulus characterizes a material's resistance to elastic deformation under shear stress, representing mechanical stiffness. Ring size distribution (RSD) quantifies the medium-range order in amorphous silica by measuring the average number of Si atoms in rings formed by the Si-O network. Detailed calculation procedures are provided in Appendix A.7.

**Inverse design metrics.**  We evaluate inverse design performance by comparing target properties with properties of generated samples using common regression metrics: Mean Absolute Error (MAE), Root Mean Square Error (RMSE), and Mean Absolute Percentage Error (MAPE). Implementation details are provided in Appendix A.8.

## 3.3 EVALUATION OF STRUCTURAL ACCURACY

To evaluate the structural accuracy of different models with different numbers of sampling steps, we perform generation using sampling steps $n_s = 1, 2, 3, 4, 5, 10, 25, 50, 100, 250$ and calculate the ADF and RDF of the generated samples. For each run, we generate the same number of samples as the training set. For the a-Si dataset, Figure 2 illustrates the RDF and ADF of samples in the training set and those generated by models with each number of sampling steps.

When generating with few sampling steps ($n_s \leq 10$), neither Material ODE nor Material SDE is able to generate structurally valid samples. Only when $n_s \geq 25$ do the RDF and ADF of the generated samples from these two models start to match the training samples. In contrast, MDShortcut is able to generate structurally accurate samples with few sampling steps, especially as evidenced by RDF where we observe a close match between the samples generated at $n_s = 1$ step and the training samples. ADF measures the angles between atomic bonds—a higher-order and more challenging metric compared to atomic distances—where we observe larger discrepancies between the generated and training samples, but MDShortcut still demonstrates advantages with few sampling steps compared to the baseline models.

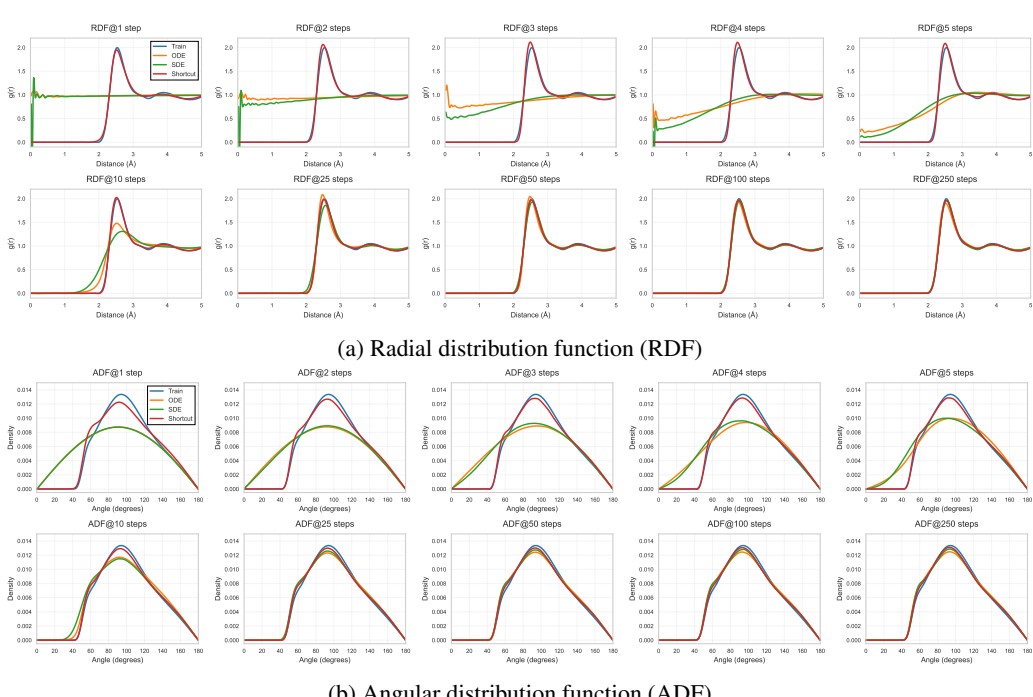

(a) Radial distribution function (RDF)

(b) Angular distribution function (ADF)

Figure 2: Structural accuracy evaluation for (a) RDF and (b) ADF of different models on the a-Si dataset with different numbers of sampling steps. ODE, SDE, and Shortcut corresponds to Material ODE, Material SDE, and MDShortcut, respectively.

We also calculate the RMSD of RDF and ADF of the generated samples in each run against the training samples, and present the RMSD versus generation time per 10,000 samples in Table 1 and Figure 5 (in Appendix A.1) to demonstrate the efficiency gain achieved by MDShortcut. The ADF RMSD of samples generated by MDShortcut with 1 step is on par with samples generated by Material ODE/SDE with 250 steps while using 1.11% of the time by ODE and 1.16% of the time by SDE. The RDF RMSD achieved by MDShortcut with 5 steps is lower than Material ODE/SDE with 250 steps while using 2.78%/2.93% of the time. These results demonstrate MDShortcut's capability to generate structurally accurate amorphous material samples with few sampling steps and low generation time. In practice, MDShortcut can generate more samples with similar or even higher structural accuracy under the same time budget compared to conventional diffusion models, facilitating the discovery of new amorphous materials where high-throughput generation is usually preferred (Liu et al., 2024).

Table 1: RMSD and generation time comparison across different models and step counts.

| $n_s$ | Material ODE | Material SDE | MDShortcut |
|---|---|---|---|
| 1 | 0.69118 / 0.00280 (1.76m) | 0.68477 / 0.00282 (1.21m) | **0.02513 / 0.00067** (2.24m) |
| 2 | 0.66442 / 0.00276 (2.24m) | 0.60972 / 0.00263 (1.96m) | **0.03922 / 0.00044** (2.89m) |
| 3 | 0.60209 / 0.00259 (2.82m) | 0.50904 / 0.00235 (2.86m) | **0.04835 / 0.00038** (3.77m) |
| 4 | 0.49544 / 0.00220 (3.71m) | 0.43276 / 0.00210 (3.80m) | **0.04653 / 0.00035** (4.70m) |
| 5 | 0.39007 / 0.00173 (4.63m) | 0.37975 / 0.00188 (3.97m) | **0.04041 / 0.00034** (5.64m) |
| 10 | 0.14917 / 0.00073 (9.10m) | 0.22054 / 0.00100 (7.92m) | **0.02244 / 0.00032** (10.06m) |
| 25 | 0.04876 / 0.00051 (22.46m) | 0.04871 / 0.00045 (20.17m) | **0.01618 / 0.00029** (24.03m) |
| 50 | 0.04203 / 0.00048 (44.92m) | 0.02631 / 0.00039 (39.79m) | **0.01487 / 0.00030** (46.90m) |
| 100 | 0.03305 / 0.00046 (1.50h) | 0.02143 / 0.00037 (1.34h) | **0.01580 / 0.00028** (1.55h) |
| 250 | 0.03353 / 0.00045 (3.38h) | 0.01905 / 0.00035 (3.21h) | **0.01548 / 0.00029** (3.59h) |

Cell format: RDF RMSD / ADF RMSD (Generation Time). **Bold**: best result per metric per $n_s$, underlined: second-best result per metric per $n_s$. RMSD values ranked by lowest value.

### 3.4 EVALUATION OF INVERSE DESIGN PERFORMANCE

We evaluate the models' inverse design performance by performing generation conditioned on specific target properties, and compare the actual properties of generated samples versus the target. We focus on evaluating two aspects of model performance: the ability to train once conditioned on all properties and generate conditioned on only the target properties, and generation with few sampling steps. Thus, we train MDShortcut and the two baseline models conditioned on all properties in the a-SiO2 dataset, denoted as suffix (All). We also train baseline models conditioned on only the target properties as a comparison, denoted as suffix (Target). The generation is always performed conditioned on only target properties, utilizing the flexible property embedding technique. Each run of generation is performed with different numbers of sampling steps $n_s \in [1, 5, 10, 25, 50, 100, 250]$. Note that the distribution of the target properties is designed to fall outside the distribution of training samples, as shown in Figure 3, to evaluate the extrapolation capability of models.

For the two properties in the a-SiO$_2$ dataset, shear modulus and ring size distribution (RSD), We perform generation conditioned on either one of them, with target shear modulus linearly interpolated between 10 and 50 [GPa] and target RSD linearly interpolated between 4 and 6 atoms, both across 2,000 samples with cubic cells with edge length 18 Å. Table 2 and Figure 4 show the divergence between the target properties and the actual properties of generated samples.

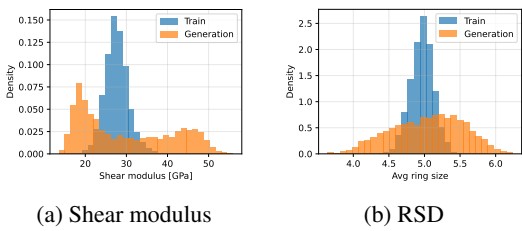

(a) Shear modulus   (b) RSD

Figure 3: Distribution comparison of (a) shear modulus and (b) RSD in the training samples versus the samples generated by MDShortcut with 100 steps.

We first observe that with the same number of sampling steps, MDShortcut consistently demonstrates performance advantage thanks to its learned shortcuts. On both properties, the alignment between the target properties and the properties of samples generated by MDShortcut with 10 steps surpasses that achieved by samples generated by all baseline models with 250 steps. Figure 6 (in Appendix A.1) further illustrates the inverse design accuracy versus generation time per 2,000 samples of different models. This highlights MDShortcut's promising capability of performing accurate inverse design with relatively low computation time resources. Note that in this case, MDShortcut's performance with 1 step exhibits some degradation compared to the structural metrics on the a-Si dataset, primarily because the flexible property embedding relies on randomization across multiple sampling steps, which is less effective with only a single step.

We also observe that MDShortcut, which is trained conditioned on the full set of relevant properties, holds its performance advantage when performing generation conditioned on a subset of target properties compared with models trained specifically conditioned on target properties. For a more fair

comparison, the conclusion holds when we compare the (All) and (Target) variants of Material ODE and Material SDE. This can alleviate the need to re-train the model, especially in scenarios where amorphous materials are coupled with a variety of properties and the flexibility of using certain subsets of all properties to perform generation is needed.

Table 2: Inverse design performance metrics comparison of different models with different numbers of sampling steps on a-SiO$_2$ dataset.

| $n_s$ | Material ODE (All) | Material ODE (Target) | Material SDE (All) | Material SDE (Target) | MDShortcut (All) |
|---|---|---|---|---|---|
| **Shear modulus [GPa]** | | | | | |
| 1 | 25.69 / 27.02 / 88.1% | 20.02 / 21.10 / 68.7% | 16.55 / 19.73 / **48.5%** | 17.75 / 20.32 / 54.8% | **15.51** / **19.30** / 53.8% |
| 5 | 19.93 / 20.75 / 70.6% | 14.28 / 14.76 / 52.0% | 13.31 / 16.30 / 39.8% | 24.34 / 26.32 / 80.5% | **3.10** / **4.10** / **16.0%** |
| 10 | 12.67 / 13.46 / 45.5% | 9.51 / 10.09 / 33.9% | 18.04 / 19.22 / 60.9% | 19.17 / 20.31 / 65.8% | **2.67** / **3.41** / **13.0%** |
| 25 | 7.29 / 8.37 / 27.2% | 6.14 / 6.96 / 20.1% | 4.73 / 5.62 / 17.2% | 3.61 / 4.35 / 14.7% | **2.86** / **3.47** / **12.7%** |
| 50 | 6.38 / 7.54 / 24.1% | 5.25 / 6.05 / 17.3% | 3.95 / 4.76 / 14.8% | 3.37 / 4.12 / 14.7% | **2.91** / **3.51** / **12.7%** |
| 100 | 5.81 / 6.92 / 22.2% | 4.82 / 5.62 / 16.0% | 3.62 / 4.40 / 13.7% | 3.43 / 4.21 / 15.2% | **2.96** / **3.55** / **12.9%** |
| 250 | 5.46 / 6.55 / 21.2% | 4.40 / 5.16 / 14.9% | 3.42 / 4.14 / 13.3% | 3.41 / 4.21 / 15.6% | **3.03** / **3.60** / **13.0%** |
| **Ring size distribution (RSD)** | | | | | |
| 1 | 2.16 / 2.26 / 44.5% | 2.08 / 2.12 / 41.2% | 2.46 / 2.49 / 49.0% | 2.18 / 2.23 / 43.2% | **1.34** / **1.36** / **26.7%** |
| 5 | 1.86 / 1.97 / 36.4% | 1.89 / 1.98 / 37.1% | 2.27 / 2.31 / 45.2% | 1.91 / 1.99 / 37.9% | **0.25** / **0.29** / **5.1%** |
| 10 | 0.85 / 0.98 / 16.7% | 0.36 / 0.44 / 7.3% | 2.41 / 2.45 / 48.0% | 2.22 / 2.28 / 43.9% | **0.15** / **0.19** / **3.0%** |
| 25 | 0.26 / 0.31 / 5.3% | 0.21 / 0.26 / 4.0% | 0.21 / 0.26 / 4.5% | 0.19 / 0.22 / 3.9% | **0.14** / **0.18** / **2.8%** |
| 50 | 0.24 / 0.28 / 5.0% | 0.21 / 0.26 / 4.2% | 0.19 / 0.24 / 4.0% | 0.18 / 0.22 / 3.8% | **0.14** / **0.18** / **2.9%** |
| 100 | 0.21 / 0.25 / 4.3% | 0.22 / 0.27 / 4.3% | 0.19 / 0.23 / 3.9% | 0.17 / 0.21 / 3.5% | **0.15** / **0.18** / **3.0%** |
| 250 | 0.18 / 0.22 / 3.8% | 0.22 / 0.28 / 4.3% | 0.18 / 0.22 / 3.8% | 0.18 / 0.21 / 3.6% | **0.15** / **0.19** / **3.0%** |

Cell format: MAE / RMSE / MAPE%. **Bold**: best result per metric per $n_s$, underlined: second-best result per metric per $n_s$. All metrics ranked by lowest value.

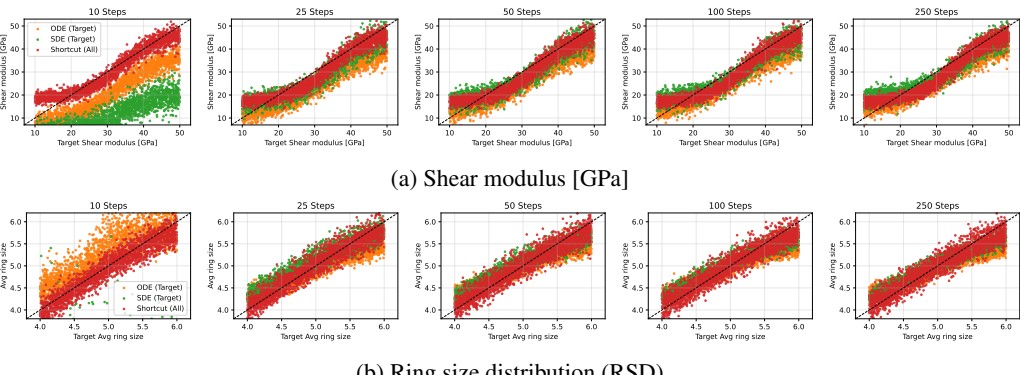

(a) Shear modulus [GPa]

(b) Ring size distribution (RSD)

Figure 4: Scatter plots of target properties versus properties of generated samples on the a-SiO$_2$ dataset comparing different models with different numbers of sampling steps for (a) shear modulus and (b) ring size distribution.

## 4 RELATED WORK

**Machine learning-aided material discovery.** Traditional material discovery involves trial-and-error approaches (Liu et al., 2017; Cai et al., 2020), where many samples are created in laboratories or simulation environments and have their properties tested. This process can be slow and resource-intensive. Some efforts propose incorporating machine learning techniques to improve the efficiency of this process. Ward et al. (2016); Sun et al. (2017); Xiong et al. (2019); Liu et al. (2020); Wang & Zhang (2021); Merchant et al. (2023); Li et al. (2025) suggest utilizing property prediction models to mitigate the need for laboratory testing or simulation of material properties. Nevertheless, such approaches require exploring large design spaces of materials, which is inherently less straightforward than inverse design approaches that directly provide the necessary atomic configurations for achieving the desired properties.

**Generative modeling of atomic systems.** Recent years have seen significant efforts on generative modeling of molecules and crystalline materials. Early efforts including Gebauer et al. (2019); Noh

et al. (2019); Court et al. (2020) are methods based on variational auto-encoders (Kingma & Welling, 2014), which are limited in effectively generating complex atomic systems (Daunhawer et al., 2022); Long et al. (2021) is a method based on generative adversarial networks (GANs) (Goodfellow et al., 2014), whose use is hampered by the unstable training of GANs (Li et al., 2018). More recent works (Wu et al., 2022; Xie et al., 2022; Hoogeboom et al., 2022; Zeni et al., 2025) based on diffusion models (Ho et al., 2020) have shown promising results. Despite these efforts, molecules and crystalline materials are inherently distinct from amorphous materials. Molecules usually are not comprised of many atoms, and crystalline materials have strong periodic atomic order that enables them to be represented by relatively few atoms in a periodic cell. In contrast, amorphous materials lack long-range atomic order and require large simulation cells to be represented, which makes generative models for molecules and crystalline materials not directly applicable to amorphous materials.

**Inverse design of amorphous materials.** Zhou et al. (2023) introduces a generative framework for predicting compositions of glass materials given desired properties. However, compositions do not provide a complete picture of atomic configurations of glasses and do not fully determine their properties (e.g., the thermal history also matters). There are a few efforts on generating atomic configurations of amorphous materials, with Comin & Lewis (2019); Xu & Hu (2023); Yong et al. (2024) based on GAN and Chen et al. (2025); Kilgour et al. (2020) based on VAE. As mentioned before, their generation quality is limited by the limitations of the underlying GAN and VAE frameworks. Kwon et al. (2024) proposes using diffusion models (Ho et al., 2020) to generate structures of amorphous carbon with desired spectroscopy, with a follow-up work for crystalline phases and grain boundaries (Lei et al., 2024), yet their exploration of broader properties and more diverse multi-element systems is limited. Yang & Schwalbe-Koda (2025); Finkler et al. (2025) expand diffusion model-based inverse design to more diverse amorphous materials, yet the challenges of subpar inference and training efficiency remain unsolved. Overall, the inverse design of amorphous materials remains underdeveloped, unlike the progress made in inverse design of other types of atomic systems.

## 5 Conclusion and Discussion

This work represents pioneering efforts toward inverse design of amorphous materials and focuses on the inference and training efficiency of generative models for amorphous materials. We introduce MDShortcut, a generative model capable of performing accurate inverse design of amorphous materials in few sampling steps. It can be trained once and perform generation conditioned on different subsets of target properties. Experiments on two amorphous material datasets provide evidence that MDShortcut achieves its design goals.

**Limitations.** Training shortcuts in MDShortcut will introduce extra computational overhead compared to Material SDE. In response, we only calculate the shortcut loss for 25% of the random batches in each epoch, keeping the additional training time manageable. Due to the random nature of diffusion model-based generative models, when generating multi-element samples, a portion of samples will not be charge balanced. Although property calculation is still possible on these samples, this can potentially hurt their usability in the real world. Considering the challenging nature of posing charge balance—a non-differentiable constraint—on generative models, this topic could be future work in its own right.

Recent efforts (Finkler et al., 2025) discover that diffusion models face inherent limitations in generating annealed structures: structures cooled down slowly and with lower energy compared to melted ones. This limitation cannot be overcome by using more sampling steps, but can be overcome by incorporating physics-guided Hamiltonian Monte Carlo refinement, with the downside of a slow sampling process. This process cannot be accelerated by learning shortcuts. Thus, solutions for improving sampling efficiency on annealed structures require further exploration.

**Use of large language models (LLMs).** The use of LLMs in this work is restricted to two aspects: 1) For proofreading purposes after the drafting of this paper is complete; 2) For writing code to convert numerical experimental results into illustration plots.

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

# A APPENDIX

## A.1 SCATTER PLOTS OF EVALUATION METRICS VERSUS GENERATION TIME

Figures 5 and 6 illustrates the structural and inverse design metrics versus generation time of different models and generation step counts. These figures provide an intuitive comparison of the inference efficiency of MDShortcut and the baseline models.

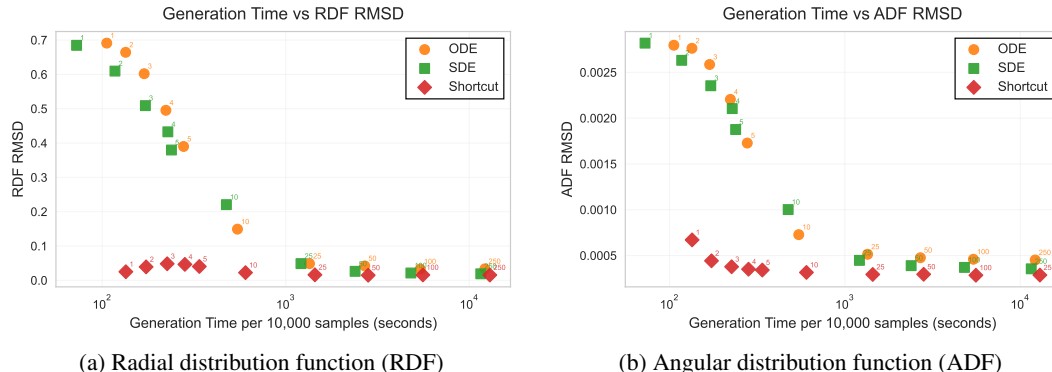

(a) Radial distribution function (RDF)     (b) Angular distribution function (ADF)

Figure 5: RMSD of (a) RDF and (b) ADF versus generation time per 10,000 samples across different models and step counts. Labels indicate the number of sampling steps for each run.

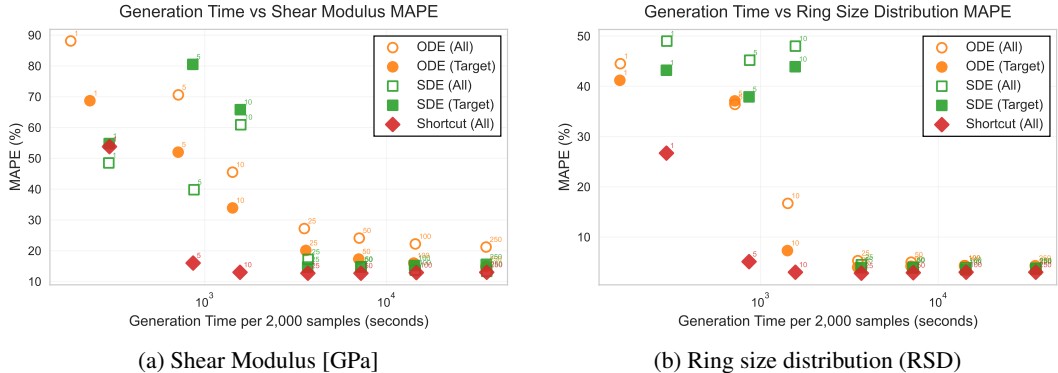

(a) Shear Modulus [GPa]     (b) Ring size distribution (RSD)

Figure 6: Property MAPE versus generation time per 2,000 samples across different models and step counts for (a) shear modulus and (b) ring size distribution. Labels indicate the number of sampling steps for each run.

## A.2 IMPLEMENTATION DETAILS OF MATERIAL ODE

Material ODE defines the ground truth $\mu$, $\sigma$, and $\mathcal{M}_t$ following the optimal transport flow (Lipman et al., 2023):

$$\mu_{\boldsymbol{X}} = \text{pbc}(\boldsymbol{X}_1 - \boldsymbol{X}_0), \ \mu_{\boldsymbol{E}} = \boldsymbol{E}_1 - \boldsymbol{E}_0, \ \sigma_{\boldsymbol{X}} = \sigma_{\boldsymbol{E}} = 0, \ \boldsymbol{X}_t = \boldsymbol{X}_0 + t\mu_{\boldsymbol{X}}, \ \boldsymbol{E}_t = \boldsymbol{E}_0 + t\mu_{\boldsymbol{E}} \quad (7)$$

where pbc denotes the periodic boundary condition, i.e., the vectors between corresponding atoms in $\mathcal{M}_0$ and $\mathcal{M}_1$ are adjusted to take the shortest path across periodic boundaries defined by $\boldsymbol{C}$.

The network $\mu_\theta$ directly predicts the positions and element embeddings components of $\mu$, denoted as $\hat{\mu}_{\boldsymbol{X}}$ and $\hat{\mu}_{\boldsymbol{E}}$, and is trained with L2 loss. Formally:

$$\hat{\mu}_{\boldsymbol{X}}, \hat{\mu}_{\boldsymbol{E}} = \mu_\theta(\mathcal{M}_t, \boldsymbol{y}, t), \quad \mathcal{L}_{\text{ODE}} = \mathbb{E}_{\mathcal{M}_0, \mathcal{M}_1, t} \left[ \|\hat{\mu}_{\boldsymbol{X}} - \mu_{\boldsymbol{X}}\|^2 + 0.5\|\hat{\mu}_{\boldsymbol{E}} - \mu_{\boldsymbol{E}}\|^2 \right] \quad (8)$$

For the random sample $\mathcal{M}_1$, we sample $\boldsymbol{X}_1$ from uniform distribution within the cell $\boldsymbol{C}$, and sample $\boldsymbol{E}_1$ from standard normal distribution.

### A.3 IMPLEMENTATION DETAILS OF MATERIAL SDE

Material SDE defines the ground truth following the score-matching SDE (Song et al., 2021) with a variance exploding noise schedule on positions and a variance preserving noise schedule with cosine progression on element embeddings. We have:

$$\mu_{\boldsymbol{X}}(\mathcal{M}_t, t) = \sigma_{\boldsymbol{X}}^2(t)\nabla_{\boldsymbol{X}}\log p(\boldsymbol{X}_t), \quad \sigma_{\boldsymbol{X}}(t) = t\sigma_{\max}^{\boldsymbol{X}}, \quad \boldsymbol{X}_t = \boldsymbol{X}_0 + \sigma_{\boldsymbol{X}}(t)\epsilon_{\boldsymbol{X}}$$

$$\mu_{\boldsymbol{E}}(\mathcal{M}_t, t) = -\frac{\pi}{2}\tan(\pi t/2)\boldsymbol{E}_t + \sigma_{\boldsymbol{E}}^2(t)\nabla_{\boldsymbol{E}}\log p(\boldsymbol{E}_t)$$

$$\sigma_{\boldsymbol{E}}(t) = \sin(\pi t/2)\sigma_{\max}^{\boldsymbol{E}} \tag{9}$$

$$\boldsymbol{E}_t = \cos(\pi t/2)\boldsymbol{E}_0 + \sigma_{\max}^{\boldsymbol{E}}\sin(\pi t/2)\epsilon_{\boldsymbol{E}}$$

where $\sigma_{\max}^{\boldsymbol{X}} = 1.7\text{Å}$ and $\sigma_{\max}^{\boldsymbol{E}} = 1.5$. $\nabla_{\boldsymbol{X}}\log p(\boldsymbol{X}_t)$ and $\nabla_{\boldsymbol{E}}\log p(\boldsymbol{E}_t)$ are the scores of positions and element embeddings, respectively. $\epsilon_{\boldsymbol{X}}$ and $\epsilon_{\boldsymbol{E}}$ are both noise sampled from standard normal distribution. Under this setting, the distribution of the random sample $\mathcal{M}_1$ will be consistent with that in material ODE.

The neural network $\mu_\theta$ is set to predict the noise of positions and element embeddings as $\hat{\epsilon}_{\boldsymbol{X}}$ and $\hat{\epsilon}_{\boldsymbol{E}}$, then the predicted scores are $-\hat{\epsilon}/\sigma(t)$. The network is trained with L2 loss on the noise, formally:

$$\hat{\epsilon}_{\boldsymbol{X}}, \hat{\epsilon}_{\boldsymbol{E}} = \mu_\theta(\mathcal{M}_t, \boldsymbol{y}, t), \quad \mathcal{L}_{\text{SDE}} = \mathbb{E}_{\epsilon, \mathcal{M}_0, t}\left[\|\hat{\epsilon}_{\boldsymbol{X}} - \epsilon_{\boldsymbol{X}}\|^2 + 0.5\|\hat{\epsilon}_{\boldsymbol{E}} - \epsilon_{\boldsymbol{E}}\|^2\right] \tag{10}$$

### A.4 IMPLEMENTATION DETAILS OF MDSHORTCUT

In our prior experiments, we find out that when calculating the shortcut loss $\mathcal{L}_{\text{SC}}$ in Eq. 4, the stochastic component in material SDE will make the training less stable, leading to suboptimal training results. To solve this, when calculating $\mathcal{L}_{\text{SC}}$ we remove the stochastic component in material SDE following the formulation of probabilistic ODE flow introduced in Song et al. (2021). Specifically, when calculating Eq. 4, $\mu$ and $\sigma$ is defined as:

$$\mu_{\boldsymbol{X}}(\mathcal{M}_t, t) = \frac{1}{2}\sigma_{\boldsymbol{X}}^2(t)\nabla_{\boldsymbol{X}}\log p(\boldsymbol{X}_t)$$

$$\mu_{\boldsymbol{E}}(\mathcal{M}_t, t) = -\frac{\pi}{4}\tan(\pi t/2)\boldsymbol{E}_t + \frac{1}{2}\sigma_{\boldsymbol{E}}^2(t)\nabla_{\boldsymbol{E}}\log p(\boldsymbol{E}_t) \tag{11}$$

$$\sigma_{\boldsymbol{X}} = \sigma_{\boldsymbol{E}} = 0$$

When a small $n_s$ number of sampling steps is used, with the Euler-Maruyama method (Eq. 2) it means a relatively large scale of noise is added to the sample at each step of the generation process. We find that this will lead to unstable generation results even with MDShortcut. Thus, when $n_s \leq 10$, we follow the probabilistic ODE flow formulation as above for generation.

The shortcut loss is calculated stochastically for only 25% of training batches. In preliminary experiments, we find that this has basically equal effectiveness compared to using 100% of batches, while significantly reducing computational overhead.

### A.5 IMPLEMENTATION DETAILS OF EGNN BACKBONE

Given an input sample $\mathcal{M} = (\boldsymbol{C}, \boldsymbol{X}, \boldsymbol{E})$, the input graph $\mathcal{G} = (\mathcal{V}, \mathcal{E})$ to the EGNN is composed of atoms in the sample as nodes in the node set $\mathcal{V}$, and each edge in the edge set $\mathcal{E}$ connects a pair of atoms with distances less than a cutoff radius of 6.5 Å. The distances are computed with periodic boundary conditions. The cutoff radius was chosen to ensures that all bonded and strongly interacting atoms share a direct edge connection while still keeping the graph sufficiently sparse.

Our EGNN implementation was composed of $L = 4$ EGNN layers. The $l$-th layer takes as input: 1) node features $\boldsymbol{H}^{(l)} \in \mathbb{R}^{n \times d_h}$ containing information of the corresponding atoms at $l$-th layer; 2) positional coordinates $\boldsymbol{X}^{(l)} \in \mathbb{R}^{n \times k \times 3}$ of the atoms, where $k = 8$ is the number of vector channels (Levy et al., 2023); and 3) edge set $\mathcal{E}$ of the graph $\mathcal{G}$, with edge attributes $\boldsymbol{e}_{ij}$ assigned to each edge.

For the initial layer, the node features $\boldsymbol{H}_i^{(0)}$ are assembled by concatenating: 1) diffusion time step $t$; 2) element embeddings: $\boldsymbol{E}_i \in \mathbb{R}^{d_E}$; 3) property embeddings: $\mathbf{h}_{y_1}, \mathbf{h}_{y_2}, \ldots, \mathbf{h}_{y_{n_p}}$ (for each property

in $\boldsymbol{y}$); and 4) time step size $\Delta t$ in the case of $u_\theta$. The positions $\boldsymbol{X}^{(0)}$ were replicated original positions $\boldsymbol{X}$ for $k$ channels. The edge attributes $\boldsymbol{e}_{ij}$ were derived from the distance embedding:

$$\boldsymbol{e}_{ij} = \tanh\left(\frac{\|\boldsymbol{X}_i - \boldsymbol{X}_j - \mathbf{o}_{ij}\|^2}{r_{\text{cut}}^2}\right) \cdot 2 - 1 \tag{12}$$

where $\mathbf{o}_{ij}$ is the offset vector accounting for periodic boundary conditions.

Each layer updated the node features and positional coordinates, incorporating self-attention (Vaswani et al., 2017) with a hidden dimension of 128 as,

$$\mathbf{m}_{ij}^{(l)} = \phi_e^{(l)}(\boldsymbol{H}_i^{(l-1)}, \boldsymbol{H}_j^{(l-1)}, \boldsymbol{e}_{ij})$$

$$\alpha_{ij}^{(l)} = \sigma(\text{MLP}_{\text{att}}(\mathbf{m}_{ij}^{(l)}))$$

$$\hat{\mathbf{m}}_{ij}^{(l)} = \alpha_{ij}^{(l)} \cdot \mathbf{m}_{ij}^{(l)}$$

$$\boldsymbol{H}_i^{(l)} = \boldsymbol{H}_i^{(l-1)} + \phi_H^{(l)}\left(\boldsymbol{H}_i^{(l-1)}, \sum_{j \in N(i)} \frac{f_{\text{cut}}(d_{ik}^{(0)}) \cdot \hat{\mathbf{m}}_{ij}^{(l)}}{n_{\text{norm}}}\right) \tag{13}$$

$$\boldsymbol{\Phi}_{ij}^{(l)} = \text{MLP}_{\text{coord}}([\boldsymbol{H}_i^{(l)}, \boldsymbol{H}_j^{(l)}, \boldsymbol{e}_{ij}]) \in \mathbb{R}^{k \times k}$$

$$\mathbf{d}_{ij}^{(l)} = \boldsymbol{X}_i^{(l-1)} - \boldsymbol{X}_j^{(l-1)} - \mathbf{o}_{ij}$$

$$\boldsymbol{X}_i^{(l)\prime} = \sum_{j \in N(i)} \frac{1}{n_{\text{norm}}} \cdot \boldsymbol{\Phi}_{ij}^{(l)} \cdot \mathbf{d}_{ij}^{(l)}$$

$$\boldsymbol{X}_i^{(l)} = \boldsymbol{X}_i^{(l-1)} + \boldsymbol{X}_i^{(l)\prime}$$

where $N(i)$ represents the neighbors of atom $i$, derived from the edge set $\mathcal{E}$ and $\sigma$ is the sigmoid activation function for self-attention. $n_{\text{norm}}$ is a normalization factor (typically proportional to the average number of neighbors) to ensure numerical stability, which we set to 40. $\phi_e^{(l)}$, $\phi_H^{(l)}$ are implemented as multi-layer perceptrons (MLPs) with SiLU activation functions and layer normalization. $\boldsymbol{\Phi}_{ij}^{(l)}$ is a learned transformation matrix that maps between the $k$ vector channels.

In our implementation, the MLPs are structured as follows,

$$\phi_e^{(l)}(\boldsymbol{H}_i, \boldsymbol{H}_j, \boldsymbol{e}_{ij}) = \text{MLP}_{\text{edge}}([\boldsymbol{H}_i, \boldsymbol{H}_j, \boldsymbol{e}_{ij}])$$

$$\phi_H^{(l)}(\boldsymbol{H}_i, \mathbf{m}_{\text{agg}}) = \text{MLP}_{\text{node}}([\boldsymbol{H}_i, \mathbf{m}_{\text{agg}}]) \tag{14}$$

And a smooth cutoff function is used to prevent discontinuities when atoms leave or enter the cutoff radius, which is defined as follows.

$$f_{\text{cut}}(r) = 2 \tanh\left(1 - \frac{\min(r, r_{\text{cut}})}{r_{\text{cut}}}\right)^2 \tag{15}$$

At the last layer, EGNN outputs $\boldsymbol{H}^{(L)}$ and $\boldsymbol{X}^{(L)}$ as the final node features and positional coordinates, respectively. We take $\boldsymbol{H}^{(L)}$ directly as the predicted element component, and the deviation between the original positions and the first channel of output positions $\boldsymbol{X} - \boldsymbol{X}^{(L,0)}$ as the predicted position component.

## A.6 STRUCTURAL METRICS CALCULATION

**Radial distribution function (RDF).** The RDF $g(r)$ quantifies the probability of finding an atom at distance $r$ from a reference atom, normalized by the corresponding probability in an ideal gas:

$$g(r) = \frac{V}{N^2} \frac{1}{4\pi r^2 \Delta r} \left\langle \sum_{i=1}^{N} \sum_{j \neq i}^{N} \delta(r - r_{ij}) \right\rangle \tag{16}$$

where $V$ is the system volume, $N$ is the number of atoms, $r_{ij}$ is the distance between atoms $i$ and $j$, and $\langle \cdot \rangle$ denotes ensemble averaging. We calculate RDF using the ASE library's `Analysis` module with a cutoff distance of 5.0 Å and 100 bins. Periodic boundary conditions are applied to ensure proper treatment of atoms near cell boundaries.

**Angular distribution function (ADF).** The ADF $P(\theta)$ measures the distribution of bond angles formed by atomic triplets. For each central atom $i$, we identify all neighbors $j$ and $k$ within a cutoff radius of 3.0 Å, then calculate the angle $\theta_{jik}$ between vectors $\mathbf{r}_{ij}$ and $\mathbf{r}_{ik}$:

$$\cos(\theta_{jik}) = \frac{\mathbf{r}_{ij} \cdot \mathbf{r}_{ik}}{|\mathbf{r}_{ij}||\mathbf{r}_{ik}|} \tag{17}$$

The distribution is binned into 180 bins covering $0°$ to $180°$ ($1°$ resolution). Neighbor lists are constructed using ASE's `NeighborList` with periodic boundary conditions.

**RMSD Calculation.** The structural accuracy is quantified using RMSD between generated and reference distributions:

$$\text{RMSD} = \sqrt{\frac{1}{N_{\text{bins}}} \sum_{i=1}^{N_{\text{bins}}} (f_i^{\text{gen}} - f_i^{\text{ref}})^2} \tag{18}$$

where $f_i$ represents the distribution value at bin $i$. For multiple samples, we first average the distributions across all structures before computing RMSD.

**Implementation Details.** Ghost atoms are excluded from all structural calculations. The RDF calculation employs the ASE library's radial distribution function method to obtain both the distribution values and corresponding distances. For the ADF calculation, we implement a custom algorithm that constructs neighbor lists for each atom, ensuring all pairwise angles are computed by considering both directions of atomic connections. The structural metrics are computed only on the actual atoms after filtering out ghost atoms, ensuring that the calculated distributions accurately represent the material structure. All calculations incorporate periodic boundary conditions to properly handle atoms near cell edges.

## A.7 PROPERTY CALCULATION

We evaluate inverse design performance using material properties relevant to the dataset. For a-SiO$_2$ samples, we focus on shear modulus and ring size distribution.

**a-SiO$_2$ Dataset Properties.** For both training and generated a-SiO$_2$ samples, properties are computed directly from atomic structures. Shear modulus is calculated using finite differences of the stress tensor: structures are relaxed with the Tersoff potential (force tolerance 0.05 eV/Å), then subjected to small strains ($\delta = 0.02$) to compute elastic constants $C_{44}$, $C_{55}$, and $C_{66}$ from stress responses. Shear modulus is their average: $G = \frac{1}{3}(C_{44} + C_{55} + C_{66})$. Ring size distribution is computed by identifying closed rings in the Si-O network using a depth-first search algorithm. Atoms are considered bonded if their distance is below 1.3 times the sum of their covalent radii (Si: 1.11 Å, O: 0.66 Å). The algorithm searches for the shortest path between bonded atom pairs while excluding the direct bond, ensuring proper ring closure with periodic boundary conditions. Ring sizes are reported as the number of Si atoms per ring, averaged across all identified rings.

**Implementation Details.** Ghost atoms are excluded from all calculations. Structural relaxations use quasi-Newton optimization with variable cell shapes, maximum 1500 steps. Ring finding incorporates periodic boundary conditions for rings crossing cell boundaries.

## A.8 INVERSE DESIGN METRICS

For inverse design evaluation, we compare target properties $y_{\text{target}}$ with computed or predicted properties $y_{\text{generated}}$ of generated samples using three standard regression metrics:

**Mean Absolute Error (MAE).**

$$\text{MAE} = \frac{1}{n} \sum_{i=1}^{n} |y_{\text{generated},i} - y_{\text{target},i}| \tag{19}$$

**Root Mean Square Error (RMSE).**

$$\text{RMSE} = \sqrt{\frac{1}{n} \sum_{i=1}^{n} (y_{\text{generated},i} - y_{\text{target},i})^2} \tag{20}$$

**Mean Absolute Percentage Error (MAPE).**

$$\text{MAPE} = \frac{100\%}{n} \sum_{i=1}^{n} \left| \frac{y_{\text{generated},i} - y_{\text{target},i}}{y_{\text{target},i}} \right| \tag{21}$$

These metrics are computed separately for each property (shear modulus, RSD, Young's modulus, Li ratio) across all generated samples. MAE provides interpretable error magnitude in original units, RMSE penalizes large deviations more heavily, and MAPE enables comparison across properties with different scales and units.

## A.9 Details on Dataset Preparation

### A.9.1 Amorphous Silicon (a-Si) Dataset

The a-Si dataset contains 10 000 samples created using LAMMPS (Thompson et al., 2022) software with the Stillinger–Weber potential (Stillinger & Weber, 1985). All molecular dynamics (MD) simulations are performed in the NPT ensemble at zero pressure. The dataset is generated by heating the crystalline silicon from 2500 K to 3000 K over 200 ps, equilibrating the melt for 300 ps, and then cooling it down again to 2500 K at a rate of $10^{12}$ K/s. The final samples are taken after equilibrating for another 300 ps at 2500 K.

### A.9.2 Amorphous Silica (a-SiO$_2$) Dataset

The a-SiO$_2$ dataset contains $6,000$ samples that share the composition of pure silica, SiO$_2$. To maximize the variation of properties between the samples generated with the same simulation workflow, relatively small unit cells are chosen with the number of atoms uniformly selected in the range of 80 to 250. Atoms are initially placed in a unit cell with a volume $V = 4 \sum_i \frac{4}{3} \pi r_i^3$, with $r_i$ being the covalent radius of the $i$-th atom, avoiding unphysical overlap between neighboring atoms. A local structure relaxation is performed on the initial configuration followed by an MD simulation in the NPT ensemble at 3500 K for 2000 ps. To limit the effects of relaxation, which we observe for our other datasets, we use an instantaneous quenching procedure by performing a local structure optimization and a subsequent equilibration at 300 K for 10 ps. All simulations are performed using LAMMPS (Thompson et al., 2022) software and the Tersoff potential parameterized by Munetoh et al. (2007).

