# OpenReview forum: "Shortcuts in Material Design: Efficient Generative Modeling of Amorphous Materials"
_ICLR.cc/2026/Conference — ICLR 2026 Conference Withdrawn Submission_

### Official Review · Reviewer_Z8v2 · 2025-10-24

**Soundness:** 1
**Presentation:** 1
**Contribution:** 1
**Rating:** 2
**Confidence:** 4

**Summary:**

This paper proposes a generative model for the generation of disordered, or amorphous, materials. Their approach is based on diffusion/flow-matching and incorporates techniques from multiple previous works. Most prominently, they use a distillation technique for one-step diffusion [1] to reduce the number of inference steps required. Additionally, they use a technique for training models conditioned on multiple variables, but enabling sampling with only a subset of them [2].

They benchmark their models with and without distillation, showing how the distillation enables generation with fewer steps, and how the conditioning techniques allows training a single model for multiple conditional tasks.

Please see more details under “strengths/weaknesses/questions”, but to summarize the reasons for my rating: while the application seems interesting and somewhat novel, and the proposed improved design seems to improve over the proposed baseline designs, I think the main problem is that the empirical evaluation is lacking comparison with other methods, and that some claimed disadvantages of previous models are not well motivated/explained, nor shown empirically. This means that the developed method is not evaluated and tested enough for a method development paper at a conference like ICLR.

[1] Kevin Frans, Danijar Hafner, Sergey Levine, and Pieter Abbeel. One step diffusion via shortcut models. In International Conference on Learning Representations, 2025
[2] Seyedmorteza Sadat, Manuel Kansy, Otmar Hilliges, and Romann M. Weber. No training, no problem: Rethinking classifier-free guidance for diffusion models. In International Conference on Learning Representations, 2025.

**Strengths:**

The use of diffusion/flow-matching for generating amorphous materials seems like a rather new and under-explored research direction application, although this is not my area of expertise.

**Weaknesses:**

The main weakness is the lack of comparison with other methods. First off, it is claimed (line 440), that amorphous materials have specific characteristics “which makes generative models for molecules and crystals not directly applicable”. I am not an expert on amorphous materials so I will refrain on weighing in on whether that is true or not, but **with a claim like that I would like to see an explanation how the method developed in this paper is specifically designed to handle these characteristics.** To me, the method seems like a somewhat straight-forward use of diffusion/flow-matching, placing it in the same category as models like CDVAE [1], DiffCSP [2], and FlowMM [3] (and I think these and/or similar methods should be mentioned as part of the related work).  **Therefore, I think that either the authors would have to clarify how their method particularly handles “the lack of long-range atomic order” and “large simulation cells”, and/or demonstrate that these baselines indeed fail at these tasks** by running the same experiments. **But even if there is an explanation why the developed method better handles this, this should still be backed up by some experiments to show that previous methods indeed fail.** It seems like a fixed cell is used (see also "Questions"), and a simple baseline would be to use a model for crystal generation, and just remove the cell-update part in training and generation and use the same procedure that you use for initializing the fixed cell. I also have some further questions about the evaluation, see “Questions”

As I interpret that the motivation of this paper is that previous methods are not applicable in this task, I expect something to **make it clear that the paper addresses these shortcomings of previous work**, but as I do not find that in the paper, I therefore do not think that the contribution is clear enough for this venue.

[1] Tian Xie, Xiang Fu, Octavian-Eugen Ganea, Regina Barzilay, and Tommi S. Jaakkola. Crystal diffusion variational autoencoder for periodic material generation. ICLR 2022
[2]  Crystal Structure Prediction by Joint Equivariant Diffusion, Rui Jiao, Wenbing Huang, Peijia Lin, Jiaqi Han, Pin Chen, Yutong Lu, Yang Liu, NeurIPS 2023
[3] Miller, B.K., Chen, R.T.Q., Sriram, A. &amp; Wood, B.M.. (2024). FlowMM: Generating Materials with Riemannian Flow Matching. ICML 2025

**Questions:**

In Figure 3, I don’t understand what you mean by “the distribution of the target properties is designed to fall outside the distribution of training “ (line 352). Do you mean that you condition on values outside of the training data? Then I don’t think a histogram is the correct way to present the results if you want to convey that the model generates according to the target: it should be a plot showing target value on one axis, and generated value on the other axis (training data of course don’t have a target value, but could perhaps be presented separately in a histogram). If just presenting a histogram, it is not possible to determine if the model generates outside of the training values because you conditioned on it, or if it is because it just generates randomly.

Line 356: what does it mean that you linearly interpolate value? That you choose which values to condition evenly spaced between these values?

Table 2: Where is MDShortcut (Target)? (also minor detail, I would prefer the “Cell format text” in tables to be put in caption for easier reading”)

As I am not particularly knowledgeable about the specific use-case: is the unit cell always kept fixed? Is this standard procedure in this application? Is this cell the same in all experiments?

Experiments: Do I interpret it correctly that the data used is from the paper Finkler et al (2025), or did you generate this yourselves?

Line 470: what do you mean with the loss for 25 % of samples?

Line 70: You say that there is a lack of trained classifiers. However, as you have data with properties that you use for training a conditional model, shouldn’t you be able to train a classifier using this data? The claim of “poor availability” on line 70 seems like unjustified in this case (as there seems to exist data to train such models)

As you use “ghost atoms” to enable a fixed number of atoms, how is this number chosen?

Minor question: When you use RMSD as a metric, I interpret this as you have a grid of distances/angles and compute RDF/ADF, for each of the values on the grid, and compute RMSD between these values calculated on the generated and reference structures. Is this the best choice, or could something like total variation distance between discrete distributions have been used?

---

### Official Review · Reviewer_ZfcL · 2025-10-27

**Soundness:** 3
**Presentation:** 1
**Contribution:** 1
**Rating:** 4
**Confidence:** 4

**Summary:**

The paper proposes learning shortcuts as an efficient training and inference procedure within diffusion models for inverse design of amorphous materials (generating materials conditioned on properties) and demonstrates generalization when conditioned on different properties. The results show improved efficiency with similar performance when compared to proposed baselines.

**Strengths:**

- The paper adapts existing ideas in diffusion models to build MDShortcut model for amorphous materials.
- The method is extremely efficient in training and inference while retaining its accuracy in materials generation and supports this with large number of experiments.
- The method shows generalization to different target properties combination than it was trained with for inverse design.

**Weaknesses:**

My main concerns can be categorized into the following:
- lack of proper contextualization with related work
- insufficient background for amorphous materials inverse design problem
- missing crucial details regarding experiment and motivation for some design choices

These weakenesses are supported in the questions below. I am willing to increase the score if the questions are adequately answered during the discussion period.

**Questions:**

- If amorphous materials "lack long-range atomic order but possess complex short- and medium-range order", how is designing local cell structures useful for downstream applications of amorphous materials? Does generating these local structures capture enough information to  predict properties? Essentially, shouldn't the task ask to design the entire material rather than one cell?
- The paper suggests that "lack of large-scale datasets and unique atomic ordering characteristics" as the main challenges for using existing crystal generation approaches [1,2,3,4]. However,
  - It is not mentioned how MDShortcut and baselines defined within the paper are trained (i.e., which datasets, its description, and if these datasets are used in previous works or not)?
  - I don't identify differences in the representation of amorphous materials (in section 2.1) and crystals. What needs to be learned differently than crystals as defined in above references? Am I missing something?
  - If one crucial difference is the number of atoms within the cell, then I am not sure which component in MDShortcut or baselines allow this to happen. The underlying architecture is inspired from EGNN which is also used in many works for crystals and molecule design.
  - From my understanding, there is nothing that could hinder using existing crystal generation methods for this task? Or, am I missing something here too?
- Regarding related work, the authors do not appropriately discuss the existing works (mentioned in the paper) on amorphous materials and in which aspects they differ. The methods proposed in these works are not considered in baselines also. This work should:
  - contrast the methodological differences across MDShortcut and these existing works
  - compare the performance of these methods with MDShortcut results
- Can MDShortcut be used for crystal generation task? What are the challenges in using this method to such a task? Please note that I am not asking for experimental results and comparison to existing works rather from conceptual lens.
- Can you point out the crucial novelty/novelties of MDShortcut? I feel the proposed learning shortcuts is a direct adaptation of self-consistency loss which has been explored in other domains. Also, how is *material differential equation* different for amorphous and crystals (which already drives most of the crystal generation works)?
- Regarding experimental setup:
  - what properties are entailed within the "All" training setting?
  - in the "target" setting, predicting and conditioning on the same target properties is not ideal (because of information leakage) and could be better to train on subset of All - {target properties} and predict the target properties. Was this setup considered?
  - why is the RMSD of RDF and ADF of generated samples compared with training samples and not test samples (which is preferred and common in crystal generation)?
  - why are the invalid generation (e.g., charge imbalanced samples) not removed?
  - what percentages of generated samples are invalid?


1. Crystal Diffusion Variational Autoencoder for Periodic Material Generation. Xie et al. ICLR 2022.
2. Crystal Structure Prediction by Joint Equivariant Diffusion. Jiao et al. NeurIPS 2023.
3. Space Group Constrained Crystal Generation. Jiao et al. ICLR 2024
4. SymmCD: Symmetry-Preserving Crystal Generation with Diffusion Models. Levy et al. ICLR 2025

---

### Official Review · Reviewer_tJ51 · 2025-10-31

**Soundness:** 2
**Presentation:** 1
**Contribution:** 1
**Rating:** 2
**Confidence:** 4

**Summary:**

The paper presents MDShortcut, an innovative probabilistic generative model aimed at accelerating and improving the inverse design of amorphous materials—such as glasses, which are structurally complex and lack long-range atomic order. The work addresses two long-standing challenges in generative modeling for materials science: slow inference due to many sampling steps and the need to train separate models for different property combinations. MDShortcut introduces an elegant solution through two key innovations. First, it learns “shortcuts” in the generative process, allowing it to produce accurate material structures in only a few sampling steps. This dramatically reduces inference time—by as much as 99% compared to existing Material ODE and SDE models—while maintaining high structural fidelity. Second, it employs a flexible material denoiser trained once on all relevant properties, which can condition generation on any arbitrary subset of target properties. This “train once, use flexibly” paradigm eliminates the need for multiple specialized models and introduces a practical mechanism using “null properties” to handle missing conditions. The model’s effectiveness is demonstrated on datasets of amorphous silicon (a-Si) and amorphous silica (a-SiO₂), where it matches or surpasses existing methods in accuracy while being significantly faster and more adaptable. These results showcase MDShortcut’s potential to make inverse material design more scalable and efficient, marking a meaningful advancement in the intersection of machine learning and computational materials science.

**Strengths:**

MDShortcut represents a major advancement in inverse material design by offering exceptional computational efficiency and flexibility compared to traditional diffusion models. It generates high-quality amorphous material structures in just a few sampling steps, drastically reducing inference time while maintaining structural accuracy. This efficiency enables high-throughput material discovery under the same computational budget that conventional models would require for far fewer samples. Despite its speed, MDShortcut preserves or surpasses the structural and inverse design accuracy of slower baselines, accurately reproducing key structural features and target properties with minimal computation. Its flexible denoiser architecture allows a single trained model to handle multiple material properties and adapt to various inference conditions without retraining, even when some conditioning properties are absent. Altogether, MDShortcut combines diffusion-level structural fidelity with orders-of-magnitude faster generation, making it a powerful and practical tool for rapid, large-scale amorphous material design.

**Weaknesses:**

Despite its remarkable efficiency, MDShortcut has several limitations related to training complexity, structural validity, and extreme-case performance. While inference is extremely fast, training the model introduces additional computational overhead due to the shortcut mechanism, though this is mitigated by applying the shortcut loss to only a subset of training batches. Structurally, MDShortcut can sometimes generate multi-element samples that are not charge balanced, reducing their physical realism and limiting direct real-world usability—a challenge that remains difficult to solve due to the non-differentiable nature of charge balance constraints. The model also inherits diffusion models’ difficulty in generating annealed, low-energy structures, as these states cannot be achieved simply by increasing sampling steps, and existing refinement methods remain computationally slow. Finally, although MDShortcut performs well structurally even with a single sampling step, its inverse design accuracy declines in such cases because its property embeddings depend on multi-step randomization. These challenges highlight areas for future work to further improve MDShortcut’s physical consistency, training efficiency, and robustness.

**Questions:**

1. [Abstract] The abstract needs to be updated, at the moment it looks vague, kindly answer the question as to what is your work precisely focusing on. Using sentences like

> MDShortcut can be trained once with all
relevant properties and perform inference conditioned on arbitrary combinations
of desired properties, mitigating the need for training one model for each combi-
nation. Experiments on two amorphous materials datasets with diverse structures
and properties demonstrate that MDShortcut achieves its design goals.

Can raise questions like : What relevant properties? Which arbitrary combinations? Which design goals?

2. [Line 48] What is the definition of "Shortcut" in your paper? Does it refers to Inverse design? Or some efficient methodology which you have implemented within inverse design? Be specific

3. [Line 55] Why is the field under-developed? Is there a particular reason, I see you mentioned unique atomic ordering, but that's not reasoning; that's a fact about amorphous material and not a reason why it is under-developed.

4. [Line 63] : Add a reference to the section/ plot where you've shown this comparison between earlier model (which you can call baseline models) and your own model. And what is SDE, ODE? Where have you defined these abbreviations?

5. [Line 65] : Conditioning on a variety of properties, which properties? And why is training efficient hindered by conditioning? It should usually become more efficient since you are now generating on a smaller and focussed subset on which you are conditioning on.

6. Have you read about implicit diffusion models (DDIM, https://arxiv.org/abs/2010.02502); how is your method different from just applying implicit diffusion model as an application?

7. [Line 95] : Again vague, what experiments? define the experiment in short and add the reference section/ plot.

8. [Line 108] : What is ghost atom? Why give it such a peculiar name? If I read it write it's just another way of either conditioning or parameterizing. Be a bit nitty about these things when you're writing in such high standard journals.

9. [Line 117 - 157] is just writing DDPM/ DDIM equations with some changed notations. Cut short all this, your work doesn't not derive anything new.

10. [Section 3] : before evaluation, tell us the training dataset and the training details please.

11 [Section 4] : Since the paper is about diffusion models for material generation, rather than focussing on papers covering fundamentals of diffusion models; you should primarily include papers in the field of material generation. I can mention a few previous submissions which I have seen previously highly similar to your work (a) Sinha, Anshuman, Shuyi Jia, and Victor Fung. "Representation-space diffusion models for generating periodic materials." arXiv preprint arXiv:2408.07213 (2024). ; (b) Luo, Youzhi, Chengkai Liu, and Shuiwang Ji. "Towards symmetry-aware generation of periodic materials." Advances in Neural Information Processing Systems 36 (2023): 53308-53329. Kindly review these works which have high similarity to your work and have been well cited previously.

12. Do you have any experimental result which we can compare with?

I feel the paper is written by a new author in the field, and as a reviewer I wish to encourage such authors. But have to maintain strict standards of ICLR platform; kindly address each of the above 12 questions and the corresponding changes meticulously so that we can make this paper and your article better.

**Details Of Ethics Concerns:**

I did not find any concerns related to Ethics

---

### Official Review · Reviewer_F3RS · 2025-11-01

**Soundness:** 2
**Presentation:** 3
**Contribution:** 2
**Rating:** 2
**Confidence:** 3

**Summary:**

This paper tackles the problem of inefficient and inflexible generative modeling for amorphous materials. To address this, the authors propose MDShortcut, a shortcut-aware denoising framework built upon a Material Differential Equation that learns step-size-dependent dynamics, enabling few-step (as few as 1–5) generation while maintaining structural accuracy. They also introduce a flexible conditional denoiser that uses null-property embeddings, allowing a single model to perform inverse design under any subset of target properties without retraining or external classifiers. Experiments on two amorphous material datasets, silicon (a-Si) and silica (a-SiO₂), show that MDShortcut achieves up to 99% reduction in inference time compared to Material SDE/ODE baselines while preserving or improving structural metrics (RDF/ADF) and inverse design accuracy for shear modulus and ring-size distribution.

**Strengths:**

1. Experimental results show that the proposed method effectively reduces the number of sampling steps needed to generate structurally accurate amorphous material samples.

2. The proposed method achieves better accuracy in inverse design tasks compared to the baseline methods.

**Weaknesses:**

1. The scope of the inverse design experiments is limited. The paper claimed the proposed method can be used for inference conditioned on arbitrary subsets of desired properties. However, the authors only conduct experiments on one dataset with two properties.

2. Missing comparison to published baselines (e.g., [1]). The authors only compare against two self-proposed baselines.

3. Insufficient evidence for training efficiency. Although the authors claim training efficiency as one contribution, they do not provide experimental results (e.g., training time of the model compared to the baselines) related to the statement.

4. In the experiments, the authors may also consider evaluating the physical properties (e.g., energetic stability) and other DFT-level validations.

[1] Zeni, Claudio, et al. "A generative model for inorganic materials design." Nature 639.8055 (2025): 624-632.

**Questions:**

The authors conduct experiments with sampling steps up to 250. In some materials generation frameworks, the number of steps can go up to 1000 (e.g., FlowMM). How does MDShortcut perform with more inference steps compared to baseline methods?

---

### Note · Authors · 2025-11-26

**Comment:**

Dear reviewers,

We appreciate your time and constructive feedback on our submission. We acknowledge that this paper requires heavy revision, thus we have decided to withdraw the manuscript to address the reviewers’ comments and prepare a revised version for a potential resubmission.

Best regards,
Authors

**Withdrawal Confirmation:**

I have read and agree with the venue's withdrawal policy on behalf of myself and my co-authors.